# Anemia among Women of Reproductive Age: An Overview of Global Burden, Trends, Determinants, and Drivers of Progress in Low- and Middle-Income Countries

**DOI:** 10.3390/nu13082745

**Published:** 2021-08-10

**Authors:** Aatekah Owais, Catherine Merritt, Christopher Lee, Zulfiqar A. Bhutta

**Affiliations:** 1SickKids Centre for Global Child Health, Toronto, ON M5G 0A4, Canada; aatekah.owais@sickkids.ca (A.O.); catherine.merritt@sickkids.ca (C.M.); christopher.lee@sickkids.ca (C.L.); 2Centre of Excellence in Women and Child Health, Aga Khan University, Karachi 74800, Pakistan

**Keywords:** women of reproductive age, anemia, hemoglobin, nutrition

## Abstract

Relatively little progress has been made in reducing anemia prevalence among women of reproductive age (WRA anemia). Interventions, policies and programs aimed at reducing WRA anemia have the potential to improve overall not only women’s, but also children’s health and nutrition outcomes. To our knowledge, this is the first review that aimed to compile evidence on the determinants and drivers of WRA anemia reduction in low- and middle-income countries (LMICs). We synthesized the available evidence on the determinants and drivers, including government policies and programs, of WRA anemia and their mitigation strategies across a wide range of countries and geographies, thus contributing to the complex and multifactorial etiology of anemia. We carried out a systematic review of published peer-reviewed and grey literature assessing national or subnational decline in WRA anemia prevalence and the associated drivers in LMICs. Among the 21 studies meeting our inclusion criteria, proximal determinants of healthcare utilization, especially during pregnancy and with the use of contraceptives, were strong drivers of WRA anemia reduction. Changes in other maternal characteristics, such as an increase in age at first pregnancy, BMI, birth spacing, and reduction in parity, were associated with modest improvements in anemia prevalence. Access to fortified foods, especially iron-fortified flour, was also a predictor of a decrease in WRA anemia. Of the intermediate determinants, an increase in household wealth, educational attainment and access to improved sanitation contributed significantly to WRA anemia reduction. Although several common determinants emerged at the proximal and intermediate levels, the set of anemia determinants and the strength of the association between each driver and WRA anemia reduction were unique in each setting included in this review. Further research is needed to provide targeted recommendations for each country and region where WRA anemia prevalence remains high.

## 1. Introduction

Anemia—a condition where the red blood cell count is reduced and the body’s ability to meet the oxygen demands of tissues is impaired—is a public health problem affecting approximately 1.76 billion people across the globe [1]. The WHO-defined hemoglobin (Hb) cut-offs, specific to age, sex and pregnancy status, are most widely used to diagnose anemia, with the threshold being <120 g/L for non-pregnant and <110 g/L for pregnant women of 15–49 years of age [2]. Anemia prevalence also varies by geographical region. Sub-Saharan Africa and South Asia have the highest anemia prevalence, and at the country level, anemia among women of reproductive age (WRA) remains a moderate-to-severe public health problem (prevalence of 20% or greater) in most WHO member states [3].

The epidemiology and etiology of anemia are multifactorial and involve a complex interplay of distal, intermediate, and proximal causes [3,4,5]. Nutrition-specific interventions, such as iron–folic acid supplementation and large-scale food fortification with iron, can improve hemoglobin/anemia status [6,7]. However, the proportion of anemia attributable to iron deficiency varies according to the underlying infection burden and other micronutrient deficiencies, especially vitamin A [8,9].

Anemia also has both immediate and more long-term consequences. Anemia, especially iron-deficiency anemia, during pregnancy can lead to not only adverse birth outcomes but also poorer cognitive development in infancy and early childhood [10,11,12]. Anemia can also have economic consequences, potentially costing countries billions of dollars in reduced productivity [4,13].

WRA are one of the groups most at risk of anemia, due to their physiological processes [3,4]. There has been little progress in reducing the anemia burden among WRA over the past two decades, with prevalence actually increasing in some South Asian and sub-Saharan African countries. Globally, anemia prevalence among both non-pregnant and pregnant WRA decreased by less than 1% per year (non-pregnant WRA: from 33% to 29%; pregnant WRA: from 43% to 38%) between 1995 and 2011 [14]. More recent estimates from WHO indicate that, globally, the prevalence of anemia among WRA has actually increased between 2011 and 2016, from 30% to 33% [15].

The 2020 adoption of anemia reduction among WRA, as an official target indicator for the second sustainable development goal (SDG2), now provides an opportunity for renewed commitment and attention to addressing this global public health challenge. Identifying effective drivers of anemia reduction, including programs and policies, and understanding how they contribute to improvements in anemia among WRA, is pivotal for creating a framework that countries can follow to improve the health and well-being of their populations.

The aim of this review was to identify studies from low- and middle-income countries (LMICs) on the drivers of national and/or subnational decline in anemia prevalence among WRA over time. In addition, we sought to identify and isolate key determinants of women’s nutrition that are directly associated with a reduction in anemia and synthesize available evidence via a narrative review. We also aimed to classify the identified determinants as distal, intermediate, or proximal, informed by the current knowledge of the epidemiology and etiology of anemia [3,4,5]. The findings of this review could assist health and nutrition policymakers to prioritize resources for achieving substantive returns on investments across a range of national and regional contexts.

## 2. Methods

### 2.1. Search Strategy

A systematic search of published peer-reviewed literature was performed to gather information on contextual factors, country interventions, policies, strategies, programs, and initiatives that may have contributed to reductions in WRA anemia over time. Two broad categories of search terms were used: WRA anemia and drivers reducing anemia. Keywords representing these terms were combined with Boolean operators and searched for in multiple databases, as follows:anaemia OR anemialow OR deficien*hemoglobin OR haemoglobin OR hb OR iron2 AND 31 OR 4driver* OR determinant* OR polic* OR program* OR intervention* OR factor* OR predictor* OR initiativ* OR strateg* OR correlat* OR supplement* OR fortif*burden OR trend* OR longitudinal OR reduc* OR trajector* OR chang* OR declin*matern* OR pregnan* OR woman* OR women* OR antenatal OR prenatal OR perinatal OR gestat*5 AND 6 AND 7 AND 8.

The search for indexed literature was conducted in nine online databases: MEDLINE, Embase, AMED, CAB Abstracts, CINAHL, Cochrane CENTRAL, LILACS, Scopus, and Web of Science. The grey literature search was conducted using Google, along with hand searches of reference lists of relevant reviews, and the direct searching of organizational websites, including the national headquarters websites for UNICEF, WHO, UNDP, WFP, FAO, the World Bank Group Open Knowledge Repository, Nutrition International, the Global Alliance for Improved Nutrition, and the International Food Policy Research Institute. The exported set of records were de-duplicated and screened for relevance in Covidence. Records were included if they met all the following criteria:Including women of reproductive age (pregnant, non-pregnant, or lactating women) from LMIC;Published in English between 1 January 1990 and 31 May 2021;Showing a reduction in anemia over two time points in national or large-scale regional studies;Examining at least one determinant or driver of anemia.

The initial indexed literature database searches returned 42,003 studies, which were reduced to 31,410 after de-duplication. Two reviewers independently performed title and abstract screening and full-text reviews. Any conflicts were resolved by a third reviewer. Applying the screening criteria to titles and abstracts left 317 records, which were then reduced to 18 upon full-text review. Grey literature and hand searches of relevant reviews identified three additional articles that met all inclusion criteria. Therefore, the total number of indexed and grey literature records that satisfied all inclusion criteria, and which are included in this review, is 21. Figure 1 summarizes the literature review process.

### 2.2. Methodological Quality Assessment

A qualitative assessment of the selected articles was performed using an appraisal tool adapted from the Newcastle–Ottawa scale for cohort studies [16], which has previously been used in other systematic literature reviews [17]. We used a number system to assess quality across three main domains: selection criteria, data analysis, and outcome measures. The theoretical score ranged from 0 (lowest grade) to 8 (highest grade). Extracted data and quality assessments were matched between two reviewers, and any disagreements were resolved through discussion with a third reviewer before reaching a consensus. We assessed the quality of evidence as being high (score above median), moderate (score equal to median) and low (score below median).

### 2.3. Data Analysis

To compare the impact of potential determinants and drivers of WRA anemia on reduction regarding its prevalence across the different studies, we estimated the compound annual growth rate (CAGR). The CAGR is an accurate way to determine the average change in values that can increase or decrease during a fixed, pre-specified interval, and has been used to assess the rate of change in the absolute prevalence of undernutrition in LMICs [18].

### 2.4. Ethics

Since this is a review of publicly available peer-reviewed and grey literature, an ethical review was not required.

## 3. Results

### 3.1. Conceptual Framework of WRA Anemia Determinants and Drivers

Our literature search identified several review articles, which we then used to create a conceptual framework to assist in identifying and interpreting determinants of WRA anemia, including secular individual- and household-level indicators, and nutrition-specific and -sensitive interventions, programs and policies (Figure 2).

### 3.2. Study Characteristics

The 21 studies included in this review [19,20,21,22,23,24,25,26,27,28,29,30,31,32,33,34,35,36,37,38,39] include data from 38 countries from Asia, Africa and Latin America (Figure 3), and assess trends in anemia among WRA between 1990 and 2017. The characteristics and quality appraisal scores of the included studies, as well as the observed annualized reduction in WRA anemia, are presented in Appendix A.

The median methodological quality score was 7 (range: 5.5–8), resulting in eight studies classified as low quality, eight studies [19,24,27,29,30,32,34,36] classified as moderate, and five studies [20,28,33,38,39] classified as high quality. Reduction in WRA anemia varied widely across the included studies, with CAGR ranging from −0.4 to −13.0%, with a median of −3.8%.

Three studies carried out regression decomposition analyses of change in WRA anemia prevalence in India (among pregnant women: [33], Tanzania (among non-pregnant women: [28] and Rwanda (all WRA: [39] using national-level data. The remaining 18 studies used various other statistical techniques (Appendix A). Five studies included only pregnant women [22,23,25,26,37], and one focused on lactating women only [31]. Results from the three studies reporting regression decomposition analyses are summarised in Table 1, while the determinants associated with anemia reduction among WRA in the other 18 studies, by regions and countries within each region, are summarized in Table 2. Effect estimates, analyzing the association between each determinant and WRA anemia in these 18 studies, are presented in Appendix A.

### 3.3. Proximal Determinants and Drivers of Anemia among WRA

#### 3.3.1. Healthcare Utilization during Pregnancy and Lactation

Healthcare utilization plays a pivotal role in women’s health and nutrition, especially during pregnancy, intra- and post-partum, and lactation. In Ethiopia, between 2005 and 2011, lactating women who breastfed for two years had significantly lower odds of being anemic, compared to women who breastfed for one year (aOR 0.76; 95% CI 0.66 to 0.87) [31]. In Zimbabwe, pregnant and lactating women had higher odds of being anemic between 2005 and 2010, compared to non-pregnant, non-lactating women (2005: aOR = 1.31, 95% CI 1.16 to 1.47; 2010: aOR = 1.23, 95% CI 1.09 to 1.34) [38]. However, this association was not observed in 2015 [38].

Antenatal care (ANC), including the receipt and consumption of iron-containing supplements, as well as deworming and malaria prophylaxis during pregnancy in endemic areas, not only result in improved birth outcomes for the mother and her child but also lead to improvements in health and nutrition for the woman in later life. Using regression-decomposition analysis, Nguyen et al. report that, of the total change in Hb levels observed among pregnant women in India, 7% was attributable to healthcare utilization (combined results presented for ANC4+, IFA 100+, deworming and weight monitoring) during pregnancy [33]. Similarly, in Ethiopia, between 2005 and 2011, Lakew et al. observed that women currently breastfeeding, who reported receiving 4 or more ANC visits during their pregnancy, had significantly lower odds of being anemic compared to women who did not receive visits (aOR 0.73; 95% CI 0.59 to 0.91; CAGR = −7.7%). In Malawi, intermittent preventative treatment during pregnancy with sulfadoxine-pyrimethamine (IPTp-SP) was associated with a decrease in anemia among pregnant women between 1997 and 2001, but not between 2002 and 2006 (1997–2001 aOR: 0.81, 95% CI 0.73 to 0.91; 2002–2006 aOR: 1.0, 95% CI 0.82 to 1.10; CAGR = −4.5%) [25]. Furthermore, in Zimbabwe, Gona et al. [38] assessed national anemia trends among WRA between 2005 and 2015, and observed that, over the ten-year period, women who did not consume iron supplements during pregnancy had consistently higher odds of anemia compared to women who did (2005: aOR = 1.17, 95% CI 1.03 to 1.33; 2010: aOR = 1.23, 95% CI 1.09 to 1.40; 2015: aOR = 1.24, 95% CI 1.08 to 1.42).

#### 3.3.2. General Healthcare Utilization

In addition to healthcare utilization during pregnancy and birth, and immediately thereafter, access to other health services, such as family planning, use of bed-nets, and deworming, are also important determinants of women’s overall health and nutrition. In Bangladesh, between 2014 and 2017, women with a history of heavy menstrual flow had significantly higher odds of being anemic compared to women who reported normal flow (aOR = 1.61, 95% CI 1.09 to 2.42) [19].

In Tanzania, using regression decomposition analysis, Heckert et al. observed that 30% of the change in anemia prevalence among WRA was attributable to contraceptive use, with an additional 8% observed change that was driven by changes in fertility rates [28]. Similarly, in Rwanda, Iruhiriye et al. reported that 43% of the observed change in anemia among women was accounted for by the use of hormonal contraceptives [39].

Lakew et al. [31] observed that in Ethiopia, between 2005 and 2011, lactating women who reported past use of contraceptives had significantly lower odds of being anemic compared to women who did not (aOR 0.68; 95% CI 0.57 to 0.80; CAGR = −7.7%). On the other hand, in India, states with higher fertility rates in 2017 experienced a greater and statistically significant reduction in anemia prevalence between 2010 and 2017 (annualized change: low SDI states = −0.98%; middle SDI states: −0.61%) [29].

In rural Vietnam, among women who participated in a weekly IFA supplementation and regular deworming program, anemia prevalence decreased from 37.8 to 14.3% (CAGR = −13.0%), and soil-transmitted helminth infections decreased from 83.7 to 13.9% between 2006 and 2012 [21]. Similarly in Malawi, non-pregnant women who lived in districts where a micronutrient and health program (MICAH) (including IFA supplementation and regular deworming) was carried out between 2000 and 2004, had significantly lower odds of being anemic compared to women who lived in control areas (OR: 0.67, 95% CI 0.57 to 0.80; CAGR = −4.7%) [30]. The MICAH program also included components for malaria control, specifically, treatment with SP and promotion of insecticide-treated bed-nets, and as mentioned above, anemia prevalence in program districts decreased significantly (OR: 0.67, 95% CI 0.57 to 0.80; CAGR = −4.7%) [30].

#### 3.3.3. Dietary Intake

Adequate dietary diversity, especially consumption of iron-rich foods is a vital determinant of the micronutrient status of individuals. Diversity in diet during the reproductive years, generally, and during pregnancy specifically, are highly correlated with improved health and nutrition outcomes for both women and their children. Using regression decomposition analysis, Nguyen et al. report that, of the total change in Hb levels observed among pregnant women in India, 1% was attributable to increased maternal meat and fish consumption [33]. In addition, among pregnant women in India, an increase of 10 mg/day/HH in iron intake, and a decrease of 100 ug/day/HH in phytate intake was found to contribute to a 10% and 1% decrease in anemia, respectively [22]. The MICAH program in Malawi also included a dietary diversity component aimed at increasing the consumption of animal-source foods and, as mentioned above, anemia prevalence in program districts decreased significantly between 2000 and 2004 (OR: 0.67, 95% CI 0.57 to 0.80; CAGR = −4.7%) [30].

Among households who participated in a homestead food production program in Bangladesh and Cambodia, egg consumption by mothers increased from one to 1.5 eggs per week (*p* < 0.05) [35]. Anemia prevalence, among mothers from communities that participated in the program in Bangladesh and Nepal, also decreased significantly between 2003 and 2006 (Bangladesh: from 51.4 to 45%, CAGR = −4.3%; Nepal: from 58 to 42.9%, CAGR = −9.6%) [35].

##### Large Scale Food Fortification Programs

National- and/or regional-level fortification policies and programs determine households’ and individuals’ access to fortified foods. Eight of the 18 included studies analyzed the impact of large-scale programs for food fortification with iron on WRA anemia [19,20,23,24,26,30,32,34]. The most common vehicle by far for iron fortification was wheat flour, followed by maize flour. Only one study reported the impact of fortified rice consumption on anemia among WRA [19]. The median CAGR for WRA anemia across these eight studies was −4.4%, ranging from −3.1 to 6.1%.

Analyzing the effectiveness of fortified rice, Ara et al. [19] observed that between 2014 and 2017, the odds of anemia among women who consumed non-fortified rice were significantly higher, compared to women who consumed fortified rice (aOR = 1.33, 95% CI: 1.01 to 1.72). Barkley et al. assessed the impact of a large-scale flour fortification program on anemia among WRA, using nationally representative data from 12 countries with fortification programs, compared to 32 countries with no programs, and found that each year of fortification decreased the odds of anemia by 2.4% (95% CI: 2.2–2.5) [20].

Chakrabarti et al. report the impact of flour fortification programs carried out in two states (Punjab and Tamil Nadu) in India, and observed that in Tamil Nadu, the decline in anemia among pregnant women over a 10-year period was 8% greater than compared to three neighboring states, which served as controls [23]. Fujimori et al. also analyzed the impact of a flour fortification program on anemia among pregnant women and observed that anemia prevalence decreased significantly between 2002 and 2008, from 25.5 to 20.2% (CAGR = −3.8%) [26].

In Cameroon, a wheat flour fortification program was also successful in reducing anemia prevalence among WRA within one year of implementation. Anemia prevalence decreased from 46.7 to 39.1% (CAGR = −5.7%), with the proportion of women having an inadequate iron intake decreasing from 85 to 66% [24]. The MICAH program in Malawi also introduced community-based fortification of maize flour and as mentioned above, WRA anemia prevalence in program districts decreased significantly (OR: 0.67, 95% CI 0.57 to 0.80; CAGR = −4.7%) [30].

In Costa Rica, following a flour fortification program, anemia among WRA declined from 18.4% in 1996 to 10.2% in 2008-09 (CAGR = −4.4%) [32]. Similarly in Fiji, following fortification of all locally milled flour in 2004, the prevalence of anemia among WRA decreased from 40.3% in 2004 to 27.6% in 2010 (CAGR = −6.1%) [34]. Reductions in ferritin deficiency (from 22.9 to 7.9%) and iron deficiency anemia (from 14.9 to 7.5%) were also observed [34].

#### 3.3.4. Maternal Characteristics

##### Maternal Age

Using regression decomposition analysis, Nguyen et al. report that, of the total change in Hb levels observed among pregnant women in India, 2% was attributable to an increase in average maternal age, which increased from 23.5 years to 29.4 years between 2006 and 2016 [33]. Similarly, Chakarbati et al. [22] observed that one additional year of age at pregnancy was associated with a 3.2% decrease in anemia prevalence, over a 10-year period.

In Guinea, WRA anemia prevalence decreased significantly between 2005 and 2012, but only among women aged 20–29 years (2005: 54.9%; 2012: 49.1%) [36]. Similarly, in Bangladesh, Ara et al. [19] found that women aged 35 years and older had higher odds of being anemic compared to their younger counterparts between 2014 and 2017 (35–45 vs. 15–25: aOR = 1.72, CI: 1.00 to 2.97; >45 vs. 15–25: 2.18, 95% CI 1.15 to 4.12). However, in Zimbabwe, age was significantly and independently associated with anemia among WRA only in 2005, but not in 2010 or 2015 (40–44 vs. 20–24 aOR = 1.66, 95% CI 1.28 to 2.16; 45–49 vs. 20–24 aOR = 1.80, 95% CI 1.35 to 2.41) [38].

In India, between 2005 and 2015, the odds of anemia among nulliparous pregnant adolescent women (15–19 years) remained consistently higher compared to older women (20–49 years), even when an overall decrease in anemia prevalence among adolescent women was observed (2005: aOR = 1.19, 95% CI 1.05 to 1.37; 2015: aOR = 1.16, 95% CI 1.03 to 1.31) [37].

##### Marital Status

In Brazil, between 2002 and 2008, pregnant women who did not have a partner had significantly higher odds of being anemic compared to women who were married or in a relationship (aOR 1.51, 95% CI 1.28 to 1.77) [26].

##### Parity and Birth Spacing

In addition to age, the number of children a woman has, as well as the interval between two consecutive pregnancies, is also directly associated with her nutritional status. Using regression decomposition analysis, Nguyen et al. report that, of the total change in Hb levels observed among pregnant women in India, 6% was attributable to the number of children under the age of five years that a woman had [33]. Having more than two children was also associated with higher odds of being anemic among pregnant women in Brazil, between 2002 and 2008 (>2 children vs. ≤2 aOR 1.61, 95% CI 1.36 to 1.91) [26].

##### Infection Burden

In Tanzania, using regression decomposition analysis, Heckert et al. observed that 14% of the change in anemia prevalence among WRA was attributable to a decrease in the prevalence of febrile episodes among young children, which the authors used as a proxy for the infection burden among WRA [28]. Similarly, in Rwanda, Iruhiriye et al. reported that 46% of the observed change in anemia among women was accounted for by a reduction in the prevalence of fever among children, aggregated at the village level [39]. In Zimbabwe, women who were HIV positive had a consistently higher odds of anemia between 2005 and 2015, compared to women without HIV (2005: aOR = 2.40, 95% CI 2.03 to 2.74; 2010: aOR = 2.35, 95% CI 1.99 to 2.77; 2015: aOR = 2.48, 95% CI 2.18 to 2.83) [38].

##### BMI

In four of the 21 included studies, and across geographical regions, the maternal BMI was significantly, and independently, associated with their anemia status [26,31,36,38]. In Brazil, between 2002 and 2008, pregnant women who were of normal weight, or overweight/obese, had significantly lower odds of being anemic, compared to women who were underweight (normal vs. underweight aOR 0.79, 95% CI 0.66 to 0.94; overweight/obese vs. underweight aOR 0.42 95% CI 0.42 to 0.66) [26]. Similarly in Ethiopia, between 2005 and 2011, lactating women with a normal weight had lower odds of suffering from anemia, compared to women who were underweight (aOR 0.78; 95% CI 0.68 to 0.89) [31]. In Guinea, between 2005 and 2012, the decrease in anemia prevalence was also statistically significant for WRA compared with normal weight (2005: 53.2%; 2012: 49.5%), and those who were overweight/obese (2005: 50.9%; 2012: 42.7%), but not for underweight WRA [36]. And in Zimbabwe, women whose BMI was > 30 had consistently lower odds of being anemic between 2005 and 2015, compared to women with normal weight (2005: aOR = 0.68, 95% CI 0.54 to 0.86; 2010: aOR = 0.61, 95% CI 0.52 to 0.72; 2015: aOR = 0.75, 95% CI 0.65 to 0.88) [38].

### 3.4. Intermediate Determinants and Drivers of Anemia among WRA

#### 3.4.1. Woman’s Education and Occupation

The association between a woman’s educational attainment and her own health and nutritional status, as well as that of her children, is well established. Using regression decomposition analysis, Nguyen et al. report that, of the total change in Hb levels observed among pregnant women in India, 24% was attributable to improvements in maternal education [33]. In Tanzania, Heckert et al. observed that 36% of the change in anemia prevalence among WRA was attributable to improvements in maternal education [28].

Over a ten-year period in India, one additional year of increase in maternal educational attainment was associated with a 1.8% decrease in anemia [22]. In Cambodia, improvement in maternal education between 2000 and 2014 was also associated with a decrease in the prevalence of anemia (adjusted β −0.08, 95% CI −0.13 to −0.03) [27]. Similarly in Ethiopia, the odds of anemia among lactating women who were employed outside the home decreased significantly between 2005 and 2011, compared to women who were not employed (aOR 0.71; 95% CI 0.63 to 0.80) [31].

#### 3.4.2. Spouse’s Education

Only one study found paternal education to be a significant predictor of anemia among WRA. Among lactating women in Ethiopia, having a husband with primary education was predictive of lower odds of anemia, compared to women whose husbands had no education (aOR 0.79; 95%CI 0.69 to 0.61) [31].

#### 3.4.3. Household Wealth

The level of income and wealth determine a household’s ability to access elements pivotal to the health and well-being of its members, including food and healthcare. Using regression-decomposition analysis, Nguyen et al. report that of the total change in Hb levels observed among pregnant women in India, 17% was attributable to a household’s socioeconomic status [33]. In Tanzania, Heckert et al. observed that 36% of the change in anemia prevalence among WRA was attributable to an increase in household wealth (results were combined for household wealth and maternal education) [28]. Similarly, in Rwanda, Iruhiriye et al. reported that 7% of the observed change in anemia among women was accounted for by the increase in the number of assets owned by the household [39].

In addition, three additional studies observed a statistically significant trend in WRA anemia reduction and an increase in household wealth. In Cambodia, the increasing wealth index was significantly associated with the observed decrease in anemia prevalence between 2000 and 2014 (adjusted β −0.13, 95% CI −0.16 to −0.11) [27]. In Ethiopia, the odds of anemia among lactating women from the middle and rich wealth tertiles decreased significantly between 2005 and 2011, compared to women from the poorest tertile (aOR 0.83; 95% CI 0.70 to 0.98) [31]. In Guinea, between 2005 and 2012, the decrease in anemia prevalence was statistically significant for WRA from the highest household wealth quintile (2005: 50.4%; 2012: 42.8%) [36]. On the other hand, in India, states with lower per capita income and education among ≥15 years old in 201, experienced a greater and statistically significant reduction in anemia prevalence between 2010 and 2017 (annualized change: low SDI states = −0.98%; middle SDI states = −0.61%) [29].

#### 3.4.4. Urban/Rural Residence

Similar to household wealth, whether a household resides in an urban or rural area has implications for the members’ ability to access elements pivotal to their health and well-being, such as access to health services, improved water and sanitation, and a lower or higher probability of exposure to malaria and soil-transmitted helminths. In Bangladesh, between 2014 and 2017, non-pregnant women living in a more rural district, Gopalganj, had significantly higher odds of being anemic, compared to non-pregnant women living in a more urban district, Gazipur (aOR: 1.67, 95%CI 1.15 to 2.41) [19]. In Zimbabwe, Gona et al. [38] observed a similar association between residence in a rural area and the odds of being anemic (2005: aOR = 1.33, 95% CI 1.08 to 1.65; 2010: aOR = 1.26, 95% CI 1.03 to 1.53). However, this association was not observed in 2015 [38].

In Cambodia, living in an urban area was significantly associated with the observed decrease in anemia prevalence between 2000 and 2014 (adjusted β −0.15, 95% CI −0.23 to −0.10) [27]. In Guinea, between 2005 and 2012, the decrease in anemia prevalence was statistically significant for WRA in two major urban areas of Conakry (2005: 54.4%; 2012: 42.6%) and Kankan (2005: 62.6%; 2012: 55.3%) [36]. In India, on the other hand, a 10% increase in urbanization between 2002 and 2012 was associated with a 2.4% increase in anemia prevalence [22].

#### 3.4.5. Water, Sanitation and Hygiene (WaSH)

The association between access to improved water, sanitation and hygiene, and nutritional status is well established. Using regression decomposition analysis, Nguyen et al. report that, of the total change in Hb levels observed among pregnant women in India, 9% was attributable to improved sanitation [33]. In Tanzania, Heckert et al. observed that 12% of the change in anemia prevalence among WRA was attributable to a decrease in the proportion of households practicing open defecation [28]. Similarly, in Rwanda, Iruhiriye et al. reported that 3% of the observed change in anemia among women was accounted for by access to improved toilets [39].

In India, Chakrabarti et al. report that a 10% reduction in the prevalence of open defecation in the community resulted in a 4.2% decrease in anemia among WRA [22]. A WaSH component was also incorporated in the MICAH program in Malawi, which included education on the construction of pit latrines and, as mentioned above, anemia prevalence in program districts decreased significantly (OR: 0.67, 95% CI 0.57 to 0.80; CAGR = −4.7%) [30].

## 4. Discussion

We aimed to identify how and to what extent trends in population levels and secular anemia determinants, as well as related programs and policies, contributed to the decrease in WRA anemia prevalence across a wide range of geographies. The prevalence of WRA anemia in LMICs is much higher compared to high-income countries. Hence, our review focused on literature from the former. Given the heterogeneity in study designs and the use of national- or regional-level surveys, we synthesized the available evidence as a narrative review, and discussed the common determinants that emerged at the proximal and intermediate level across the settings included in this review. None of the included studies reported on distal determinants of anemia among WRA.

Among the proximal determinants, an increase in healthcare utilization, including antenatal care and iron supplementation during pregnancy, and access to contraceptives and deworming, appeared to be most strongly associated with a decrease in WRA anemia. In malaria-endemic countries, the improved use of bed-nets and IPTp-SP was also a strong predictor of WRA anemia reduction.

The use of contraceptives reduces the risk of anemia, likely through a decrease in menstrual bleeding [40] and adverse birth outcomes, via increased inter-pregnancy intervals and/or decreased parity [41]. The relationship between contraceptive use and the reduction in anemia among WRA was also observed in countries where the burden of disease increased. In Nepal, for example, even though anemia prevalence increased from 35% in 2006 to 40% in 2016, the use of hormonal contraceptives was associated with a decrease in the odds of anemia compared to not using any contraception [42].

Improved dietary intake, especially an increase in consumption of iron-rich foods, also predicted improvements in WRA anemia prevalence. The most significant improvements were observed in countries and regions that instituted large-scale food fortification initiatives, where the average annualized rate of reduction in WRA anemia was 4.4%. Overall, improvements in dietary diversity, especially increased consumption of animal-source foods, were associated with more modest decreases in anemia among WRA. This is not surprising, since the proportion of anemia attributable to iron deficiency varies across countries and regions.

The relationship between the consumption of iron-rich foods and the prevalence of anemia is also observed in settings where the burden of disease increased between two time points. In India, anemia prevalence among WRA increased between 1998 and 1999 (52%) and 2005–06 (56%) [43]. However, the higher consumption of coarse cereals, which are high in iron, was significantly associated with lower anemia prevalence [43].

Proximal determinants that also contributed significantly to the anemia burden included maternal characteristics related to reproductive health. Pregnancy and breastfeeding are known risk factors for anemia, so it is not surprising that, in Zimbabwe, the odds of anemia among pregnant or lactating women were higher in 2005 and 2010 [38]. What was surprising was that this association was not observed in 2015. This can perhaps be explained by the significant increase in the consumption of iron-containing supplements during pregnancy in this period. Between 2005 and 2015, iron-containing supplement consumption increased from less than 25% in 2005 to greater than 40% in 2015 [38]. This indicates that iron supplementation during pregnancy is a key intervention for reducing the prevalence of anemia, not only among pregnant women but also in the postpartum period and perhaps beyond, especially in countries with a high burden of iron deficiency.

Surprisingly, breastfeeding for two years was observed as being protective against anemia in Ethiopia [31]. This relationship was also observed among WRA in rural Western China, where anemia prevalence actually increased from 34% in 2001 to 46% in 2005 [44]. This is likely due to lactational amenorrhea when iron losses due to breastfeeding are likely to be much lower than during menstruation.

Higher parity and short birth intervals are also known risk factors for anemia among WRA. Results from regression decomposition analysis from India found that, among pregnant women, a decrease in the number of children < 5 years accounted for 6% of the increase in measured Hb, observed between 2006 and 2016 [33]. Higher parity (>4 children) was also associated with the increased odds of anemia among women in Nepal in 2016, where anemia prevalence actually increased between 2006 and 2016 [42]. On the other hand, having more than one child was associated with a decrease in the odds of anemia in rural Western China in 2001 and 2005 [44]. One possible explanation for this contradictory finding could be China’s family planning program during this time, which allowed for more than one child based on sub-population characteristics, such as socioeconomic conditions [45]. Therefore, it is possible that those families who chose to have more than one child belonged to a higher socioeconomic stratum.

The proximal maternal characteristics of BMI and age at pregnancy were also associated with observed changes in anemia prevalence. In all four studies that assessed the relationship between maternal BMI and anemia, women who were underweight (BMI < 18.5 kg/m^2^) had consistently higher odds of anemia compared to women who had a BMI higher than 18.5 kg/m^2^, even when the overall prevalence of anemia among women of reproductive age decreased over time [26,31,36,38]. Being overweight or obese was also associated with decreased odds of anemia compared to being of normal weight in Nepal in 2016, even when the overall prevalence of anemia increased across all BMI groups between 2006 and 2016 [42].

Similarly, increased maternal age at pregnancy was associated with a significant decline in anemia in India [22,33], and adolescent pregnancy was consistently associated with higher odds of anemia compared to pregnant women of 20–49 years, over a ten-year period [37]. These findings suggest that interventions aimed at improving overall maternal nutrition, such as balanced energy protein supplementation, and reducing the rates of adolescent pregnancies could be effective in decreasing the prevalence of anemia among women of reproductive age.

Of the determinants classified as intermediate in this review, an increase in household wealth and maternal educational attainment were most strongly associated with a decrease in WRA anemia prevalence. Regression decomposition analyses in India and Tanzania revealed that 17–36% of the change in WRA anemia can be explained by changes in household wealth and maternal education [28,33]. Higher educational attainment was also protective against anemia in India, where an increase in anemia prevalence was observed between 1998 and 2006 (52 to 56%) [43]. Increased access to improved sanitation, and a reduction in the proportion of the population practicing open defecation, were also identified as strong drivers of improvements in anemia prevalence among women of reproductive age.

On the other hand, even though higher socioeconomic status was associated with lower odds of anemia in rural Western China in 2001, by 2005, the odds of anemia among women from the highest and lowest wealth indexes were similar [44]. Similarly, in Nepal, the rate of increase in anemia prevalence was 0.12% per year among women from households in the highest wealth quintile, compared to 0.01% per year among women from households in the lowest wealth quintile [42]. However, Nepal has instituted several successful programs and policies, aimed at increased access to improved toilets and the cultivation of kitchen gardens among poorer households, and several interventions aimed at increasing consumption of iron-rich foods, which could have resulted in improved nutritional status among WRA from the poorest households [42].

Residence in urban areas was also found to be protective against the risk of anemia. The only exception was observed in India, where increased urbanization was associated with an increase in anemia prevalence between 2002 and 2012 [22]. One explanation for this contradictory finding is the rapid pace of urbanization in the country over the past several years, with a corresponding increase in populations living in urban slums who have reduced access to healthcare services and improved WaSH [46].

### Limitations

Since we only included studies using observational data to assess the trends in WRA anemia prevalence and its determinants and drivers, it is not possible to infer causality. We also only found three studies that carried out regression decomposition analysis, considered to be one of the most comprehensive and rigorous statistical methods for analyzing observational data. However, the fact that several common determinants emerged, across the countries, geographies, and types of statistical analysis included in this review, strengthens our findings and conclusions. Additionally, even though we searched a wide range of indexed and grey literature databases, the possibility of an incomplete retrieval of eligible studies remains. Furthermore, our results are only applicable to LMICs. The drivers and determinants of anemia among WRA in high-income countries are likely to be different or even in contradiction to our findings.

## 5. Conclusions

Although there has been little progress in reducing the anemia burden globally among women of reproductive age over the past two decades, some countries have made substantial improvements. Our review identifies a common set of determinants and drivers across geographies, with varying anemia epidemiology and etiology. Determinants associated with greater improvements in WRA anemia prevalence include healthcare utilization and access to fortified foods, as well as improved household wealth and maternal educational attainment. Future research should aim to capture more comprehensive information on the country-specific etiology of anemia among WRA. More comprehensive and harmonized data collection would enable a comparison of the disease burden across countries and geographies, as well as providing targeted recommendations for each country and region where the prevalence of anemia among WRA remains high.

## Figures and Tables

**Figure 1 nutrients-13-02745-f001:**
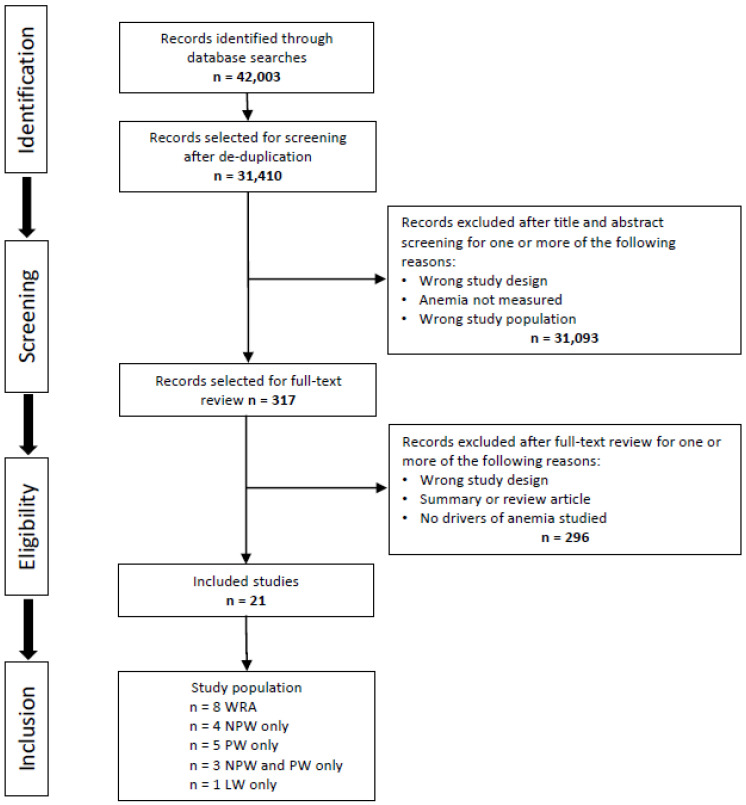
Flow diagram of the literature review process. WRA: women of reproductive age; NPW: non-pregnant women; PW: = pregnant women; LW: lactating women.

**Figure 2 nutrients-13-02745-f002:**
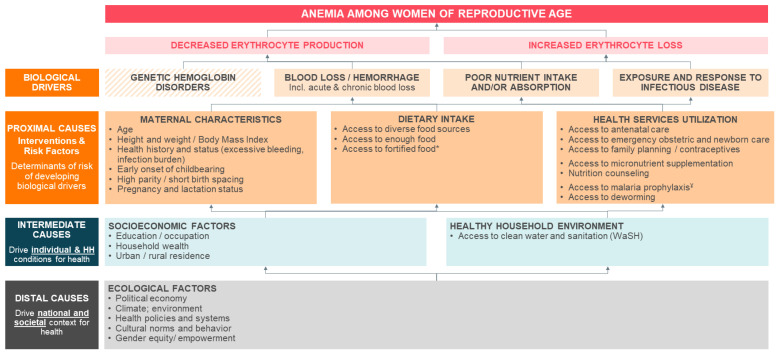
Conceptual framework of maternal anemia determinants. Determinants include those identified during the systematic review process. * Including from large-scale food fortification programs. ^¥^ Including insecticide-treated mosquito nets (ITN) and intermittent preventive treatment of malaria in pregnancy (IPTp).

**Figure 3 nutrients-13-02745-f003:**
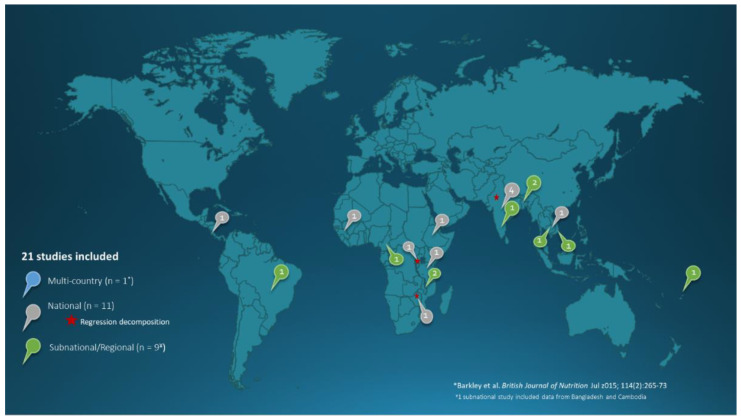
Setting and sample representativeness of included studies.

**Table 1 nutrients-13-02745-t001:** Summary of changes in anemia prevalence, statistically explained by changes in anemia determinant indicators within regression decomposition analyses.

			Nguyen et al. [33]	Heckert et al. [28]	Iruhiriye et al. [39]
Causes	Category	Determinant/Indicator	India (2006–2016)	Tanzania (2005–2010)	Rwanda (2005–2010)
Proximal	Healthcare Utilization	ANC	7% *		
		IFA consumption	7% *		
		Deworming	7% *		
		Contraceptive use		30%	43%
		Change in fertility		8%	
		Nutrition counseling	7% *		
	Dietary intake	Dietary diversity	1%		
	Maternal characteristics	Age	2%		
		Birth spacing	6%		
		Infection burden		14%	46% ^¥^
Intermediate	Socioeconomic factors	Household wealth	17%	36% **	7%
		Improved sanitation	9%	12%	3%
		Maternal education	24%	36% **	

ANC: antenatal care; IFA: iron/folic acid; * combined results presented for ANC4+, IFA 100+, deworming and weight monitoring; ** combined results reported for household wealth and maternal education; ^¥^ village-level aggregate of fever in children, used as a proxy for village-level fever-inducing infections.

**Table 2 nutrients-13-02745-t002:** Summary of determinants associated with anemia reduction among WRA, across regions and studies included.

Region	Country	Author, Year	Time Period	Determinants Associated with Decrease in Maternal Anemia
Nationally representative studies				
East Asia and Pacific	Cambodia	Greffeuille, 2016	2000 to 2014	SOCIOECONOMIC FACTORS: Household wealth, urban residence and maternal education
Latin America and Caribbean	Costa Rica	Martorell, 2015	1996 to 2008–09	DIETARY INTAKE: National flour fortification program
South Asia	India	Bellizzi, 2020	2005 to 2015	MATERNAL CHARACTERISTICS: Age at pregnancy > 20 years
	India	Chakrabarti, 2018	2002–04 to 2012–13	MATERNAL CHARACTERISTICS: Age DIETARY INTAKE: Dietary diversity SOCIOECONOMIC FACTORS: Urbanization, maternal education HEALTHY HOUSEHOLD: Decrease in open defecation
	India	Swaminathan, 2019	2010–2017 ^¥^	SOCIOECONOMIC FACTORS: Socio-demographic index * Low SDI states: −0.98% (−1.35 to −0.60) Middle SDI states: −0.61% (−0.97 to −0.22) High SDI states: −0.21% (−0.60 to 0.25)
Sub-Saharan Africa	Zimbabwe	Gona, 2021	2005 to 2015	MATERNAL CHARACTERISTICS: Age < 40, BMI > 30, not pregnant/lactating, HIV negative HEALTH SERVICES UTILIZATION: IFA in pregnancy SOCIOECONOMIC FACTORS: Urban residence
	Ethiopia	Lakew, 2015	2005 to 2011	MATERNAL CHARACTERISTICS: BMI > 18.5, breastfeeding for 2 years HEALTH SERVICES UTILIZATION: 4+ ANC, contraceptive use SOCIOECONOMIC FACTORS: Household wealth, maternal education and occupation
	Guinea	Wirth, 2019	2005 to 2012	MATERNAL CHARACTERISTICS: Maternal age 20–29 years, and > normal BMI SOCIOECONOMIC FACTORS: Household wealth, urban residence
Multi-region	Multi-country	Barkley, 2015	Various	DIETARY INTAKE: National flour fortification programs
Subnational/Regional studies				
East Asia and Pacific	Fiji	Shultz, 2012	2004 to 2010	DIETARY INTAKE: National flour fortification program
	Viet Nam	Casey, 2017	2005 to 2012	HEALTH SERVICES UTILIZATION: Weekly IFA and regular deworming program
Latin America and Caribbean	Brazil	Fujimori, 2011	2002 to 2008	MATERNAL CHARACTERISTICS: Parity, normal and higher BMI DIETARY INTAKE: National flour fortification program SOCIOECONOMIC FACTORS: Marital status
South Asia	Bangladesh	Ara, 2019	2014–15 to 2016–17	MATERNAL CHARACTERISTICS: Maternal age > 35, self-reported history of heavy menstrual flow DIETARY INTAKE: Fortified rice consumption SOCIOECONOMIC FACTORS: Urban residence
	India	Chakrabarti, 2019	2002-04 to 2012-13	DIETARY INTAKE: State-level flour fortification program
Sub-Saharan Africa	Cameroon	Engle-Stone, 2017	2009 to 2012	DIETARY INTAKE: Large-scale wheat flour fortification
	Malawi	Feng, 2010	1997 to 2006	HEALTH SERVICES UTILIZATION: Malaria prevention (IPTp and bed-net use) during pregnancy
	Malawi	Kalimbira, 2010	2000 to 2004	Micronutrient and health (MICAH) program, incl. DIETARY INTAKE: Flour fortification, weekly IFA, dietary diversification HEALTH SERVICES UTILIZATION: Malaria prevention and regular deworming HEALTHY HOUSEHOLD: Improved WaSH
Multi-region	Bangladesh and Cambodia	Talukder, 2014	BD: 2003 to 2006 Cam: 2005 to 2007	DIETARY INTAKE: Homestead food production program (dietary diversity)

ANC: antenatal care; IFA: iron/folic acid; IPTp: intermittent preventive treatment of malaria in pregnancy; WaSH: water, sanitation and hygiene. Table legend: Proximal causes, intermediate causes. ^¥^ used GBD 2017; * composite indicator of development status, with numbers being annualized percentage change and corresponding 95% confidence interval in anemia prevalence.

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
