# Peer review of "Anemia among Women of Reproductive Age: An Overview of Global Burden, Trends, Determinants, and Drivers of Progress in Low- and Middle-Income Countries"

_nutrients, 2021, doi:10.3390/nu13082745_

Round 1

Reviewer 1 Report

Very well written comprehensive review revealing new connections among wide known reasons of anemia among women in reproductive age around the globe.

On the basis of their experience gained during the work the authors could suggest or even point e.g. in the Discussion section the best approach how to design future studies in this field for both governmental bodies and research institutions.

Such instructions could help comparing prospective results from different countries and universalize targeted recommendations.

Author Response

Comments

Response

Reviewer 1

Very well written comprehensive review revealing new connections among wide known reasons of anemia among women in reproductive age around the globe.

General comment. No response required.

On the basis of their experience gained during the work the authors could suggest or even point e.g. in the Discussion section the best approach how to design future studies in this field for both governmental bodies and research institutions. Such instructions could help comparing prospective results from different countries and universalize targeted recommendations.

Added is Conclusions (lines 532 – 535)

Reviewer 2 Report

The work from Owais and colleagues is a systematic review of published peer-reviewed and grey literature focused on studies from low- and middle-income countries, to assess the decline in women of reproductive age anemia (WRA anemia) prevalence and the associated drivers.

Reducing WRA anemia is a global challenge, and its multifactorial etiology and multiple causes makes it difficult to recognize specific strategies for intervention. Therefore, reviewing evidences on determinants, drivers and mitigation strategies of WRA anemia, can provide an important view for improving women health and well-being in different settings.

This review is well done. The methodology is complete, including a precise description of the search strategy, quality assessment and data analysis.

However, a few issues should be addressed before publication:

The review focuses on low- and middle-income countries, which is reasonable giving the multiple variables involved in anemia. However, this should be mentioned both in the title and in the abstract, representing a limit in the feasibility of results in relation to developed countries. The exclusively inclusion of low- and middle- income countries and the exclusion of developed ones, should also be underlined in the discussion and in the conclusion.

The reason of the choice to include only low- and middle-income countries should also be presented in the discussion.

The exclusion of developed countries also contributed to reported results, that need to be discussed and interpreted based on that.

For example, maternal high BMI has been associated to lower odds of anemia. This should be deeper commented (lines 477-485), as on the contrary in developed countries overweight and obese women have been shown to be at higher risk of anemia, representing a consequence of malnutrition.

In general, industrialized countries might present characteristics, such as a strong urbanization and consequent unhealthy lifestyles, that may drive to different results compared to low- and middle-income countries.

The discussion should be remodeled taking into account the specific social and healthcare characteristics of the included populations. A final comment of possible differences occurring in a developed country is suggested.

Moreover, the discussion needs some further improvement:

  • lines 435-39. The possible reason why the use of contraceptive is associated with a reduction of anemia might be speculated.
  • Lines474-5. “On the other hand, having more than one child was associated with a decrease in odds of anemia in rural Western China in 2001 and 2005”. The contradiction with previously presented results in not discussed.
  • Results reported at lines 405-406 (“In India on the other hand, a 10% increase in urbanization between 2002 and 2012 was associated with a 2.4% increase in anemia prevalence”) should also be discussed. Indeed, these results appear to be in contrast with those obtained in other low- medium-income countries, where urbanization is associated with a decrease in anemia prevalence. May the degree and the features of urbanization affect the reported different results?

Minor comments:

Lines 75-6. “We also aimed to classify the identified determinants as distal, intermediate or proximal”: actually, distal determinants are not reported in results.

Acronyms should be explained all along the text and in figure/table legends.

Figure 1: in the legend, NPW, PW, LW should be explained

Figure 2: in the legend, HH (blue and light blue squares), EmONC (orange square), ITN and IPTs (legend) should be explained

Table 1 and 2: again, acronyms such as ANC and IFA should be explained in the legend

Line 190: explain ANC

Author Response

Comments

Response

Reviewer 2

The work from Owais and colleagues is a systematic review of published peer-reviewed and grey literature focused on studies from low- and middle-income countries, to assess the decline in women of reproductive age anemia (WRA anemia) prevalence and the associated drivers.

Reducing WRA anemia is a global challenge, and its multifactorial etiology and multiple causes makes it difficult to recognize specific strategies for intervention. Therefore, reviewing evidences on determinants, drivers and mitigation strategies of WRA anemia, can provide an important view for improving women health and well-being in different settings.

This review is well done. The methodology is complete, including a precise description of the search strategy, quality assessment and data analysis.

However, a few issues should be addressed before publication.

General comment. No response required.

The review focuses on low- and middle-income countries, which is reasonable giving the multiple variables involved in anemia. However, this should be mentioned both in the title and in the abstract, representing a limit in the feasibility of results in relation to developed countries. The exclusively inclusion of low- and middle- income countries and the exclusion of developed ones, should also be underlined in the discussion and in the conclusion.

The reason of the choice to include only low- and middle-income countries should also be presented in the discussion. 

Title has been revised and the focus on low- and middle-income countries (LMICs) has been specified in Abstract.

This is has been added in Discussion (lines 426 – 429).

The exclusion of developed countries also contributed to reported results, that need to be discussed and interpreted based on that.

a)     For example, maternal high BMI has been associated to lower odds of anemia. This should be deeper commented (lines 477-485), as on the contrary in developed countries overweight and obese women have been shown to be at higher risk of anemia, representing a consequence of malnutrition.

b)     In general, industrialized countries might present characteristics, such as a strong urbanization and consequent unhealthy lifestyles that may drive to different results compared to low- and middle-income countries.

Since our review explicitly focuses on LMICs, we don’t feel a discussion of the drivers and determinants of WRA anemia in high-income countries is necessary. However, to address this comment we have included this as a limitation of our study in Discussion (lines 524 – 526).

The discussion should be remodeled taking into account the specific social and healthcare characteristics of the included populations. A final comment of possible differences occurring in a developed country is suggested.

Many of the countries included in our review share similar social and healthcare characteristic, especially those in Sub-Saharan Africa and South Asia. Therefore, in our opinion, a reframing of the Discussion to take these characteristics into account is not needed.

We have included the possibility of our results not being applicable to high-income countries as a limitation in Discussion (lines 524 – 526).

Moreover, the discussion needs some further improvement:

a)     Lines 435-39. The possible reason why the use of contraceptive is associated with a reduction of anemia might be speculated.

b)     Lines474-5. “On the other hand, having more than one child was associated with a decrease in odds of anemia in rural Western China in 2001 and 2005”. The contradiction with previously presented results in not discussed.

c)     Results reported at lines 405-406 (“In India on the other hand, a 10% increase in urbanization between 2002 and 2012 was associated with a 2.4% increase in anemia prevalence”) should also be discussed. Indeed, these results appear to be in contrast with those obtained in other low- medium-income countries, where urbanization is associated with a decrease in anemia prevalence. May the degree and the features of urbanization affect the reported different results?

a.      This has been added in Discussion (lines 474 – 476)

b.     This has been added in Discussion (lines 521 – 526)

c.      This has been added in Discussion (lines 565 – 571)

Minor comments: 

Lines 75-6. “We also aimed to classify the identified determinants as distal, intermediate or proximal”: actually, distal determinants are not reported in results.

None of the included studies reported any distal determinants. A clarifying sentence has been included in Discussion (lines 432 – 433).

Acronyms should be explained all along the text and in figure/table legends.

This has been done

Figure 1: in the legend, NPW, PW, LW should be explained

This has been done

Figure 2: in the legend, HH (blue and light blue squares), EmONC (orange square), ITN and IPTs (legend) should be explained

This has been done

Table 1 and 2: again, acronyms such as ANC and IFA should be explained in the legend

This has been done

 Line 190: explain ANC

This has been done